:ᵖ: PLOS | ONE

# Impaired cardiac performance, protein synthesis, and mitochondrial function in tumor-bearing mice

Taylor E. Berent[1], Jessica M. Dorschner[1], Thomas Meyer[1], Theodore A. Craig[1], Xuewei Wang[2], Hawley Kunz[3], Aminah Jatoi[4], Ian R. Lanza[3], Horng Chen[5], Rajiv Kumar (ID)[1,3,6]*

1 Division of Nephrology and Hypertension, Department of Medicine, Mayo Clinic, Rochester, Minnesota, United States of America, 2 Division of Biomedical Statistics and Informatics, Department of Health Sciences Research, Mayo Clinic, Rochester, Minnesota, United States of America, 3 Division of Endocrinology, Department of Medicine, Mayo Clinic, Rochester, Minnesota, United States of America, 4 Department of Medical Oncology, Mayo Clinic, Rochester, Minnesota, United States of America, 5 Department of Cardiovascular Diseases, Mayo Clinic, Rochester, Minnesota, United States of America, 6 Department of Biochemistry and Molecular Biology; Mayo Clinic, Rochester, Minnesota, United States of America

* rkumar@mayo.edu

**Data Availability Statement:** Sequencing data was deposited in the Gene Expression Omnibus database (https://www.ncbi.nlm.nih.gov/geo/; GEO # GSE130704).

## Abstract

### Background

To understand the underlying mechanisms of cardiac dysfunction in cancer, we examined cardiac function, protein synthesis, mitochondrial function and gene expression in a model of heart failure in mice injected with Lewis lung carcinoma (LLC1) cells.

### Experimental design

Seven week-old C57BL/J6 male and female mice were injected with LLC1 cells or vehicle. Cardiac ejection fraction, ventricular wall and septal thickness were reduced in male, but not female, tumor-bearing mice compared to vehicle-injected control mice. Cardiac protein synthesis was reduced in tumor-bearing male mice compared to control mice (p = 0.025). Aspect ratio and form factor of cardiac mitochondria from the tumor-bearing mice were increased compared control mice (p = 0.042 and p = 0.0032, respectively) indicating a more fused mitochondrial network in the hearts of tumor-bearing mice. In cultured cardiomyocytes maximal oxygen consumption and mitochondrial reserve capacity were reduced in cells exposed to tumor cell-conditioned medium compared to non-conditioned medium (p = 0.0059, p = 0.0010). Whole transcriptome sequencing of cardiac ventricular muscle from tumor-bearing vs. control mice showed altered expression of 1648 RNA transcripts with a false discovery rate of less than 0.05. Of these, 54 RNA transcripts were reduced ≤ 0.5 fold, and 3 RNA transcripts were increased by ≥1.5-fold in tumor-bearing mouse heart compared to control. Notably, the expression of mRNAs for apelin (*Apln*), the apelin receptor (*Aplnr*), the N-myc proto-oncogene, early growth protein (*Egr1*), and the transcription factor *Sox9* were reduced by >50%, whereas the mRNA for growth arrest and DNA-damage-inducible,

**Funding:** This work was supported by a grant from the Fred C. and Katherine B. Andersen Foundation. Dr. Kumar's and Dr. Jatoi's programs are supported NIH grants DK107870 (RK) and R01CA195473 (AJ). The funders had no role in study design, data collection and analysis, decision to publish, or preparation of the manuscript.

**Competing interests:** The authors have declared that no competing interests exist.

beta (*Gadd45b*) is increased >2-fold, in ventricular tissue from tumor-bearing mice compared to control mice.

## Conclusions

Lung tumor cells induce heart failure in male mice in association with reduced protein synthesis, mitochondrial function, and the expression of the mRNAs for inotropic and growth factors. These data provide new mechanistic insights into cancer-associated heart failure that may help unlock treatment options for this condition.

## Introduction

Understanding the mechanisms underlying cardiac failure in the context of cancer has the potential to reduce morbidity and chemotherapy-associated cardiac toxicity. Humans with cachexia associated with lung, pancreatic and gastrointestinal cancers not only demonstrate skeletal muscle wasting, but also manifest cardiac atrophy and reduced cardiac function[1–10]. Cramer et al. showed that patients with colorectal cancer had reduced left ventricular (LV) ejection fraction, peak oxygen consumption, and higher levels of the endothelium-derived C-terminal-pro-endothelin-1, independent of chemotherapy[2]. Pavo et al. found elevated serum concentrations of NT-proBNP, MR-proANP, MR-proADM, CT-pro-ET-1 and hsTnT in an unselected population of patients with cancer prior to induction of cardiotoxic anticancer therapy. The biomarkers and copeptin were related to all-cause mortality, suggesting the presence of myocardial damage is directly linked to disease progression. Chemotherapy results in additional cardiac dysfunction and heart failure in some cancer patients[11, 12]. There is a need to understand the mechanisms of cardiac dysfunction in the context of chemotherapy-naive cancer patients in order to develop therapies that can mitigate cancer associated heart failure and identify those at increased risk of chemotherapy-induced heart failure.

Pre-clinical studies where tumor cells are administered to rodents have been developed to define the effects of tumor cells on cardiac function[3]. In AH-130 hepatoma Wistar-Han rats, loss of body weight in tumor-bearing rats is accompanied by a reduction in heart weight, LV mass and contractility, increased fibrosis, decreased calcium-dependent ATPase activity, and altered heart muscle redox balance[13–15]. Wistar-Han rats injected with Walker-256 tumor cells also present with cardiac failure[16]. Further, cardiac fractional shortening and cardiac wall thickness were significantly reduced in male CD2F1 mice administered colon-26 adenocarcinoma cells compared to control mice[17]. Tumor-bearing mice also exhibited increased cardiac fibrosis, decreased cardiac myofibrillar proteins, increased protein ubiquitination, and altered composition of MHC. Increased expression of brain natriuretic peptide, c-Fos and decreased peroxisome proliferator-activated receptor α and its responsive gene carnitine palmitoyltransferase 1 are noted in tumor-bearing mice[17, 18]. The expression of biomarkers of protein degradation is increased in the hearts of female CD2F1 mice with systolic and diastolic dysfunction induced by colon-26 tumor cells[19]. Others have found heart failure, decreased cardiac cell size, increased autophagy and normal caspase 3 activity and protein ubiquitination in male (but not female) CD2F1 (Balb/c X DBA2) mice injected with colon-26 adenocarcinoma cells[18, 20, 21]. In sum, animal models demonstrate reduced cardiac function, increased cardiac fibrosis with variable evidence for altered protein ubiquitination. There is very limited information describing changes in protein synthesis, mitochondrial function, and RNA expression.

In a mouse model of lung cancer cell-associated cachexia, we now describe novel mechanisms that are associated with cardiac atrophy and impaired function. We show that cardiac protein synthesis is markedly reduced in the absence of overt evidence for increased protein degradation. We demonstrate morphological and biochemical alterations in cardiac mitochondria in tumor-bearing mice and attenuated mitochondrial oxygen uptake in cultured human cardiomyocytes exposed to conditioned media from tumor cells. Marked reduction in the mRNA expression of apelin and the apelin receptor were evident in hearts of tumor-bearing mice. The described model of lung cancer cell-induced cardiac cachexia is likely to be useful in the isolation of factors causing reduced cardiac function in the context of cancer, and may open the door to therapies designed to reduce cardiac failure in patients with cancer.

## Materials and methods

### Ethics statement

The Mayo Clinic Animal Care and Use Committee specifically approved animal experiments conducted in this study: Mayo Clinic IACUC protocol number A00002917-17.

### Cardiomyocyte and LLC1 tumor cell culture

Lewis lung carcinoma (LLC1) cells (American Type Culture Collection) were grown in Dulbecco's Modified Eagle's Medium (DMEM) with 10% fetal bovine serum (FBS). AC16 human cardiomyocytes (Millipore Sigma) were grown in DMEM/F12 with 12.5% FBS. Cells were grown at 37° C with 5% $CO_2$. Conditioned medium (CM) from LLC1 cells grown for 24h was filtered and diluted to 25% in fresh media prior to treatment of AC16 cardiomyocytes.

### Animal use

Experiments were approved by the Mayo Clinic Institutional Animal Care and Use Committee. Seven week-old C57BL/J6 male and female mice were obtained from the Jackson Laboratory and maintained on PicoLab® 5053 rodent diet. 0.1 mL of PBS or $1 \times 10^6$ LLC1 tumor cells in PBS were injected subcutaneously into the left flank of 7 week-old mice. Body mass was recorded prior to and 14 days post-injection. At 14 days, the animals were euthanized with carbon dioxide and tissues were isolated, weighed and stored at -80° C.

### Echocardiography

14 days post vehicle- or LLC1 tumor-injected mice were anesthetized with isoflurane (1.5% v/v in $O_2$) before echocardiography[22, 23]. A Vivid 7 Ultrasound Unit (GE Medical Systems) was used for the echocardiography[22, 23]. All the measurements and calculations were performed according to the American Society of Echocardiography guidelines. EchoPAC software (GE) was used for data analysis.

### Fractional protein synthesis rates

Mice were injected intraperitoneally with 1.5 mmoles/kg $^{13}C_6$-phenylalanine (Cambridge Isotope Laboratories, 99 atom percent excess) 25 mins prior to euthanasia and collection of heart tissue[24]. Cardiac ventricular tissue was isolated and frozen in liquid nitrogen. 20 mg of pulverized heart tissue was suspended in 5% sulfosalicylic acid and sonicated for 10 mins at 4° C to extract the tissue fluid (TFF) free amino acids. The tissue pellets were washed with petroleum ether and heated to 60° C for 90 mins in 0.8 M NaOH. Tissue supernatants obtained after NaOH treatments were hydrolyzed at 110° C for 20h in 3N hydrochloric acid. The hydrolysates and TFF samples were purified on cation exchange columns (BioRad, AG 50W-X8

resin) and dried. The TFF and hydrolysate amino acids were analyzed via mass spectrometry as described previously[25]. The fractional synthesis rates (FSR) of cardiac muscle protein were calculated from isotopic enrichment of $^{13}C_6$ phenylalanine in tissue (Ie), using the TFF phenylalanine enrichment (Pe) as the precursor pool[24] and the equation: FSR (%/h) = [Ie/ (Pe * t)] * 100, where t represents time[26].

## Cardiac RNA isolation and mRNA sequencing

Ventricular tissue was suspended in Qiazol® Lysis Reagent (Qiagen) and homogenized. Total RNA purification was performed using a miRNeasy® Mini Kit (Qiagen) and quantified via Nanodrop™ ND-1000 spectrophotometry. 200 ng of RNA/sample were submitted to Mayo's Gene Expression Core for Illumina library preparation (RNA sample kit, v2) and sequencing (150 base paired ends index reads, Illumina HiSeq4000 platform) as previously described[27, 28]. Mouse genome build (mm10) was used in sequencing read alignment. Sequencing data was deposited in the Gene Expression Omnibus database (https://www.ncbi.nlm.nih.gov/geo/; GEO # GSE130704).

## Analysis of mRNA sequencing data

mRNA-seq data was processed by the Bioinformatics Core to identify mRNAs that were differentially expressed between the PBS-injected and LLC1 tumor cell-injected groups as previously described[27, 28].

## Transmission electron microscopy of cardiac mitochondria

Ventricular samples were stored in Trump's fixative (4% paraformaldehyde, 1% glutaraldehyde, 100 mM cacodylic acid, 2 mM $MgCl_2$) at 4°C. The Mayo Electron Microscopy Core processed the samples for osmium tetraoxide staining, resin embedding and ultrathin-sectioning for transmission electron microscopy (TEM)[28] with a JEOL JEM-140 microscope with an accelerating voltage of 80 kV. Longitudinal section images were collected at random with a fixed magnification of 20,000. Three samples from tumor-bearing and three samples from control mice were processed for TEM. Three images of each sample were analyzed. Mitochondria were traced using ImageJ software to provide cross-sectional area, aspect ratio and form factor data[29, 30].

## AC16 cardiomyocyte oxygen consumption rates

Mitochondrial oxygen consumption was measured with a Seahorse XF24 Analyzer (Agilent Technologies) as previously described[31]. AC16 cells were grown to confluency and treated for 24h with either 25% (v/v) LLC1 CM or 25% non-CM. Oxygen consumption rates were measured following sequential addition of 1 μM oligomycin, 1 μM FCCP and 0.5 μM rotenone + antimycin A and normalized to protein.

## Immunoblot analysis

Cardiac muscle homogenates were prepared by snap freezing samples in liquid nitrogen, followed by pulverization and suspension of the powder in T-PER (Thermo). Sodium dodecyl sulfate polyacrylamide gel electrophoresis of cardiac tissue homogenates was performed using NuPage 4–12% gradient gels (Invitrogen) with loading of 2 μg protein per lane. Following electrophoresis, proteins were transferred to polyvinylidene fluoride membranes, followed by treatment with antibodies directed against components of mitochondrial OXPHOS complexes (abcam, ab 110413) and developed using a chemiluminescent reagent kit (Roche). Corrections

for protein loading were performed using SYPRO Ruby (Thermo). The density of various protein bands was quantitative using Image J (NIH).

## Histological assessment of fibrosis in cardiac tissue sections

Cardiac tissues derived from tumor-bearing mice or vehicle-injected mice were embedded in paraffin, fixed, sectioned and stained with hematoxylin-eosin or Masson trichrome stain. Sections were assessed by a pathologist for the presence or absence of fibrosis.

## Statistical methods

Statistical differences between vehicle- and LLC1 tumor-bearing conditions were analyzed using Student's two-tailed t test, assuming equal variance. A p value of $< 0.05$ was regarded as statistically significant. A cut-off for the false discovery rate was set at 0.05 to determine significant changes in mRNA expression.

# Results

## Echocardiography reveals impaired heart function in male but not female LLC1 tumor-bearing mice

14 days post LLC1 tumor cell or PBS vehicle injection, heart echocardiography showed that tumor-bearing male mice had significantly thinner LV posterior walls (LVPW, $0.628 \pm 0.04$ mm systole and $0.482 \pm 0.03$ mm diastole) compared to vehicle-injected mice ($1.122 \pm 0.04$ mm systole and $0.698 \pm 0.01$ mm diastole; both $p < 0.0001$, Fig 1A and 1B). The interventricular septa (IVS) of the tumor-bearing male mice in systole were also significantly thinner, $0.833 \pm 0.04$ mm compared to $1.099 \pm 0.02$ mm in the vehicle-injected mice ($p < 0.0001$, Fig 1C). The IVS in diastole was not significantly different between the two groups ($p = 0.29$, Fig 1D). The LV internal dimensions (LVID) of the tumor-bearing males were significantly increased, $3.303 \pm 0.08$ mm systole and $4.113 \pm 0.04$ mm diastole, compared to $2.607 \pm 0.08$ mm systole and $3.699 \pm 0.11$ mm diastole in vehicle-injected mice ($p < 0.0001$ and $p = 0.0035$, Fig 1E and 1F). Heart ejection fraction (EF) was significantly impaired in tumor-bearing male mice, $46.56 \pm 2.5$ (Teichholz) and $48.22 \pm 2.5\%$ (Cubed) versus $63.4 \pm 1.1$ (Teichholz) and $65.1 \pm 1.1\%$ (Cubed) in vehicle-injected male mice (both $p < 0.0001$, Fig 1G and 1H). The percent fractional shortening (% FS) was significantly impaired in tumor-bearing male mice, $19.67 \pm 1.3\%$ compared to $29.4 \pm 0.8\%$ in vehicle-injected male mice ($p < 0.0001$, Fig 1I).

Female mice injected with tumor cells displayed no differences on echocardiography compared with PBS-injected mice (Fig 2, panels A-I).

## Cardiac fractional protein synthesis rate is decreased in male tumor-bearing mice

We investigated whether decreased protein synthesis might be contributing to the cardiac atrophy in LLC1 tumor-bearing male mice. Cardiac ventricular samples were analyzed to assess incorporation of labeled $^{13}C_6$-phenylalanine into protein[26]. The fractional protein synthesis rate was significantly decreased in tumor-bearing male mice compared to PBS-injected mice (FSR $0.386 \pm 0.02\%$/h in tumor-bearing mice versus $0.492 \pm 0.03\%$/h in vehicle-injected mice ($p = 0.025$, Fig 3A)). These data are consistent with the measured cardiac mass in tumor bearing and control mice. Fourteen days following the injection of tumor cells or vehicle, heart masses tended to be lower in tumor-bearing male mice ($110.7 \pm 3.3$ g in tumor-bearing mice compared to $121.9 \pm 5.4$ g in vehicle-injected mice, $p = 0.10$, Fig 3B). Tumor-bearing male mice had heart/body mass ratios of $0.0047 \pm 0.0001$ compared to $0.0052 \pm 0.0002$

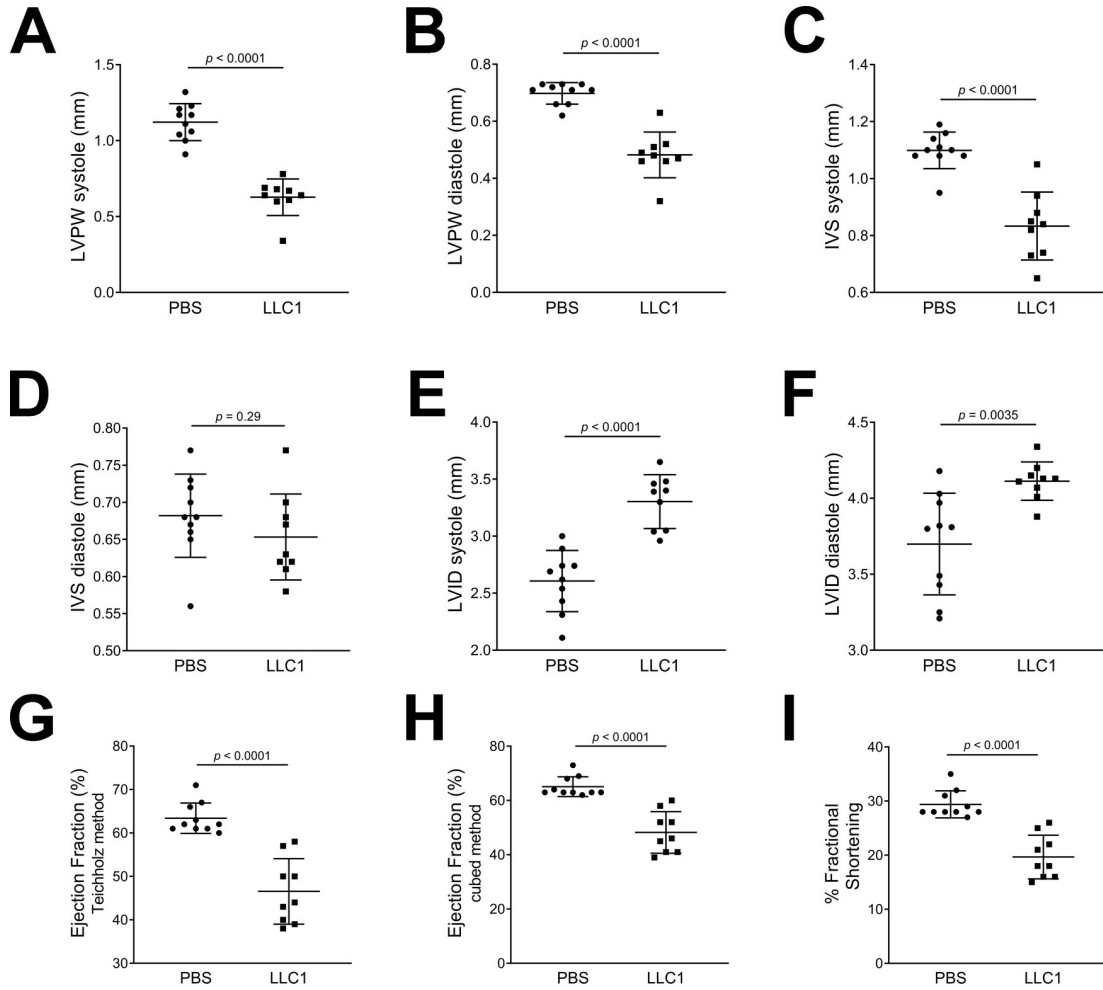

**Fig 1. Effects of PBS- and LLC1 tumor cell-injection on ventricular echocardiography in male mice 14 days post-injection of vehicle or cells.** A. LV posterior wall (LVPW) thickness in systole. B. LVPW thickness in diastole. C. Interventricular septum (IVS) thickness in systole. D. IVS thickness in diastole. E. LV internal dimension (LVID) in systole. F. LVID in diastole. G. Ejection fraction (%) calculated by the Teichholz method. H. Ejection fraction (%) calculated by the cubed method. I. Percentage (%) fractional shortening.

in the vehicle-injected group (p = 0.10, Fig 3C). In female mice the heart masses were similar in PBS-injected and tumor-bearing groups (93.3 ± 2.5 g and 96.1 ± 1.9 g, respectively (p = 0.38)).

## Altered morphology of cardiac mitochondria of male LLC1 tumor-bearing mice

Mitochondria play a major role in the generation of energy in cardiac tissue[32]. We assessed changes in mitochondrial morphology using transmission electron microscopy of cardiac muscle from tumor-bearing mice and controls. Cardiac intra-myofibrillar mitochondria from the tumor-bearing mice appeared altered in appearance (Fig 4A) compared to cardiac mitochondria of control mice. The ventricular mitochondrial cross-sectional area was 0.3889 ± 0.028 μm$^2$ in tumor-bearing mice compared to 0.4821 ± 0.042 in the PBS-injected group (p = 0.084, Fig 4B). Cardiac mitochondrial number in tumor-bearing mice was 31.22 ± 2.6 mitochondria/image compared to 26.56 ± 1.9 in the PBS group (p = 0.17, Fig 4C).

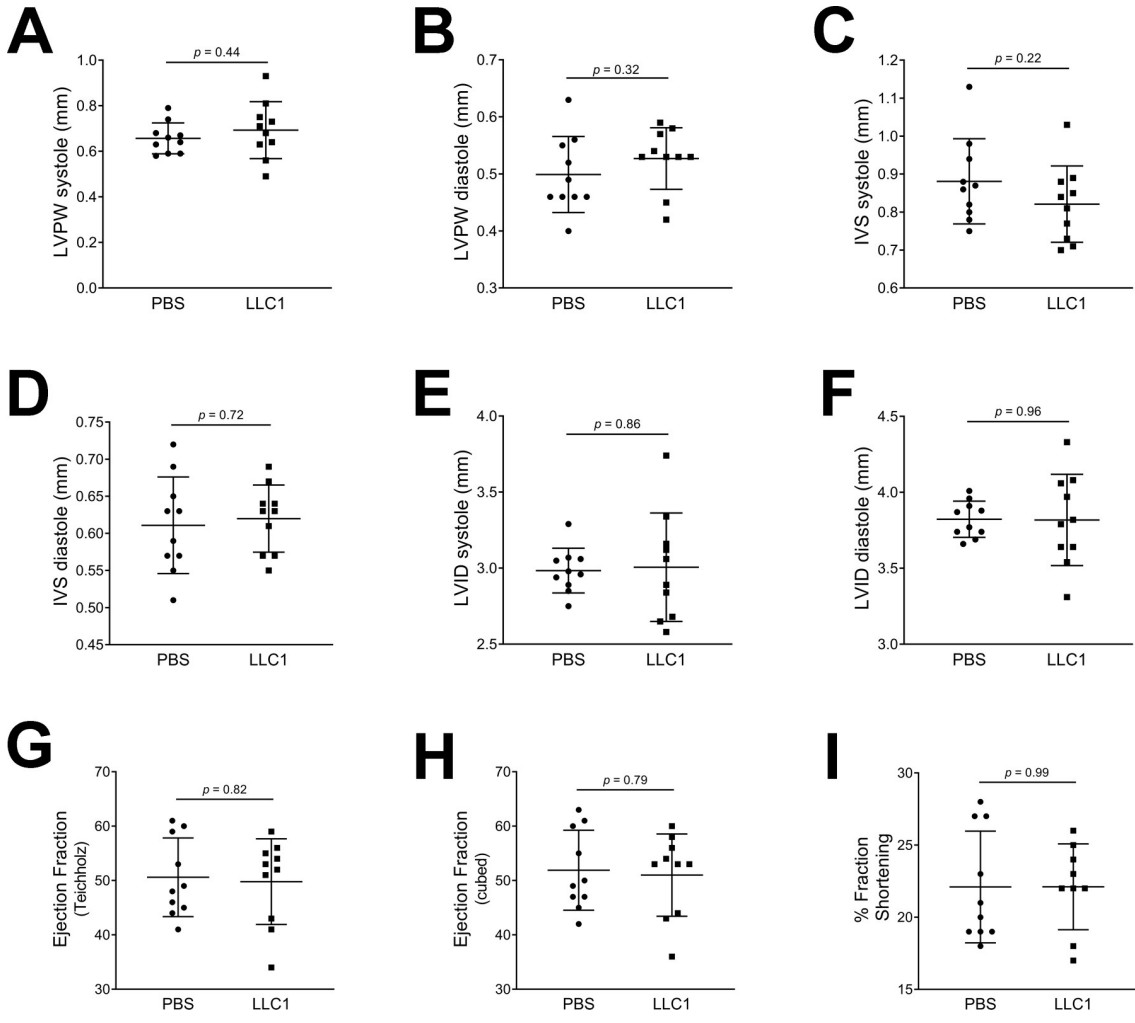

**Fig 2. Echocardiography in PBS- and LLC1 cell-injected female mice.** A. LVPW thickness in systole. B. LVPW thickness in diastole. C. IVS thickness in systole. D. IVS thickness in diastole. E. LVID in systole. F. LVID in diastole. G. Ejection fraction (%) calculated by the Teichholz method. H. Ejection fraction (%) calculated by the cubed method. I. Percentage (%) fractional shortening.

Mitochondrial density was statistically similar between the groups (Fig 4D). Cardiac mitochondrial aspect ratio (AR, major/minor axis) was increased in tumor-bearing mice, 1.764 ± 0.05 compared to 1.636 ± 0.03 in the vehicle-injected group, p = 0.042, revealing that

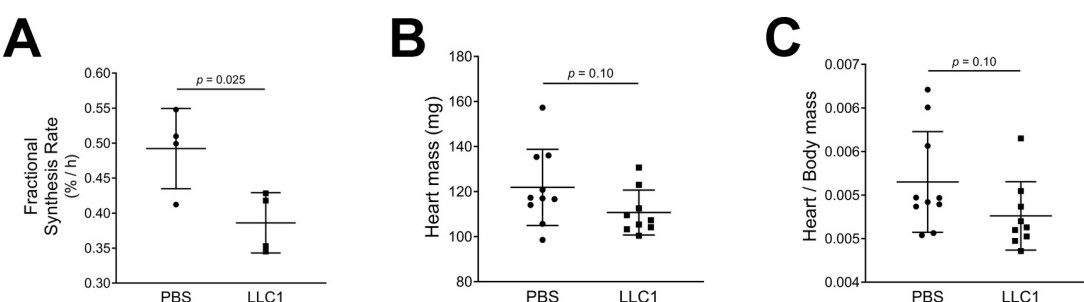

**Fig 3.** A. Protein fractional synthesis rates in the hearts of tumor-bearing and control mice. B. Heart masses of tumor-bearing and control mice. C. Heart/body mass ratios of tumor-bearing and control mice.

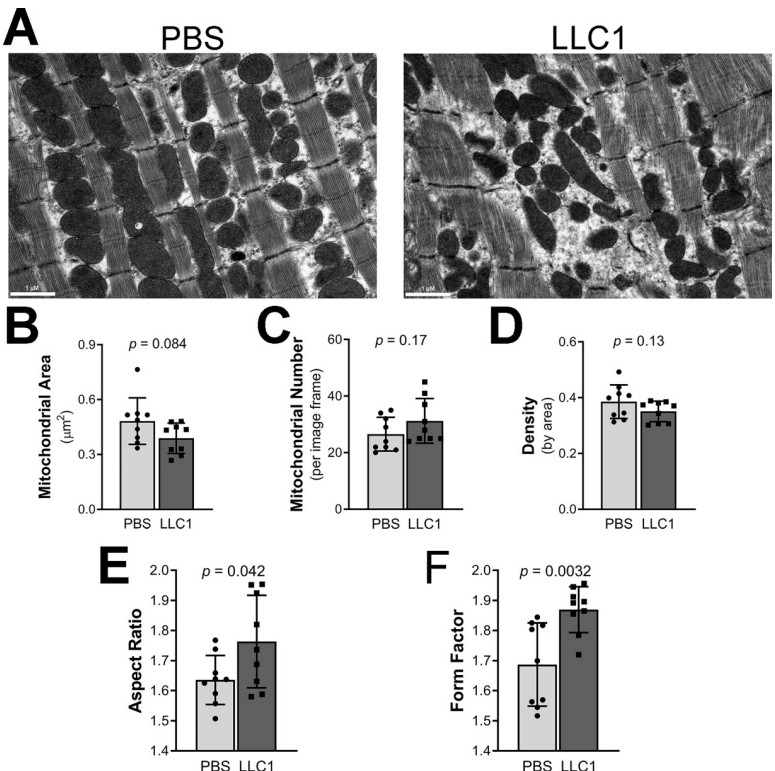

**Fig 4. Effects of LLC1 tumor cells on cardiac myofibrillar mitochondrial morphology.** A. Ventricular tissue from PBS-injected mice (left panel) and LLC1 tumor-bearing mice (right panel). B. Average cross-sectional area ($\mu m^2$) of intramyofibrillar mitochondria. C. Mitochondrial number per image. D. Mitochondrial density by area, calculated per image frame area of 43.588 $\mu m^2$. E. Mitochondrial aspect ratio. F. Mitochondrial form factor (FF).

cardiac mitochondria have a more elongated shape in tumor-bearing mice (Fig 4E). Form factor (FF, perimeter$^2$/4π·area) was increased in cardiac tissue from tumor-bearing mice (FF 1.869 ± 0.03 in tumor-bearing mice compared to 1.687 ± 0.05 in the vehicle-injected group, p = 0.0032 (Fig 4F) indicating more branched mitochondria.

## LLC1 tumor cell-conditioned media impairs maximal mitochondrial oxygen consumption in cultured human cardiomyocytes

Since in vivo evidence points to altered mitochondrial morphology in tumor-bearing mice, we tested the effects of LLC1 tumor cell conditioned medium on cardiac mitochondrial oxygen consumption rates (OCR) in cardiomyocytes maintained in culture. Monolayers of cultured human cardiomyocytes (AC16 cells) were exposed to fresh cardiomyocyte media + 25% LLC1 tumor cell CM or LLC1 culture medium not exposed to cells (non-CM) for 24 hours prior to OCR measurements. Basal oxygen consumption rates were unchanged in AC16 cells exposed to CM or non-CM (CM OCR 9.07 ± 0.3 pmol $O_2$/min/$\mu g$ protein and non-CM OCR 9.19 ± 0.6 pmol $O_2$/min/$\mu g$ protein, p = 0.85, Fig 5A and 5B). After adding oligomycin, which blocks mitochondrial complex V, the OCR of both CM and non-CM treated-wells was similar (Fig 5A). Maximal OCR of cardiomyocytes in the presence of the uncoupling agent FCCP was significantly lower in the cells exposed to the LLC1 CM compared to the non-CM (CM OCR 12.58 ± 0.7 pmol $O_2$/min/$\mu g$ protein versus non-CM OCR16.67 ± 1.1 pmol $O_2$/min/$\mu g$ protein, p = 0.0059, Fig 5A and 5C). Mitochondrial reserve capacity was reduced in the cardiomyocytes

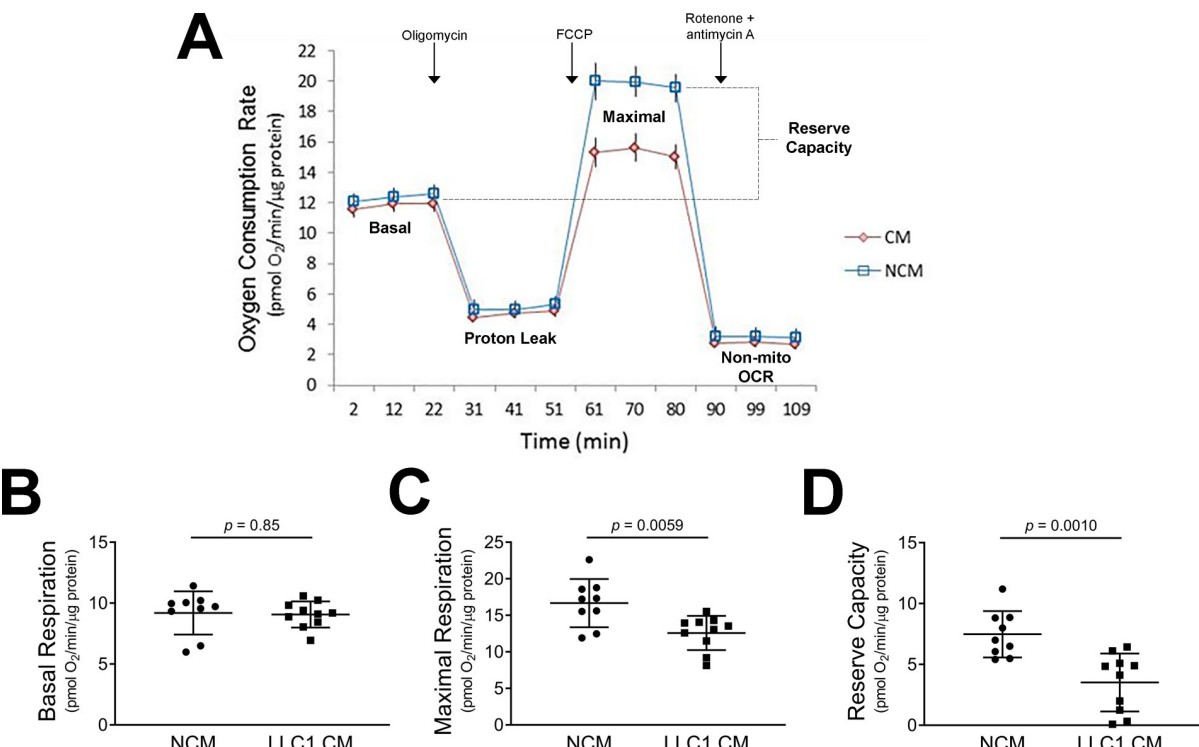

**Fig 5. LLC1 tumor cell conditioned medium impairs maximal and reserve mitochondrial oxygen consumption rates in cardiomyocytes.**
A. Effects of LLC1 tumor cell conditioned media (CM) and non-conditioned media (NCM) on AC16 human cardiomyocyte oxygen consumption rates, normalized to total μg protein. B. Cardiomyocyte basal respiration. C. Cardiomyocyte maximal respiration. D. Cardiomyocyte mitochondrial reserve capacity.

exposed to CM versus non-CM (CM OCR 3.51 ± 0.8 pmol $O_2$/min/μg protein and non-CM OCR 7.48 ± 0.6 pmol $O_2$/min/μg protein, p = 0.0010, Fig 5D).

## Assessment of mitochondrial OXPHOS complex proteins by immunoblot

To determine whether changes in mitochondrial morphology and mitochondrial oxygen consumption induced by tumor cells, are associated with reductions in mitochondrial OXPHOS protein expression, we carried out immunoblot analysis of cardiac tissues obtained from tumor-injected or vehicle-injected mice (Fig 6). Complex II, III, IV and V were reduced in cardiac tissues derived from tumor-injected mice (P < 0.0072, 0.0049, 0.029, and 0.027, respectively) compared to vehicle-injected mice.

## Mitochondrial respiratory complex mRNA transcript levels in cardiac tissue of male tumor-bearing mice

Mitochondrial respiratory complex mRNA transcripts were quantified by Illumina RNA-seq. Small increases (1.3 fold or lower) in the expression of mRNAs for Complex I—NADH dehydrogenase (ndufa1, ndufa3, ndufa6, ndufa7, ndufa11, ndufa13, ndufs7, ndufv1, ndufb8), Complex II—succinate dehydrogenase (sdhb), Complex III—ubiquinol-cytochrome c reductase (uqcr10, uqcr11), Complex IV—cytochrome c oxidase (cox6a2, cox8b) and Complex V—ATP synthase (atp5d) were observed in tumor-bearing mice compared to controls (FDR < 0.05, Table A in S1 File).

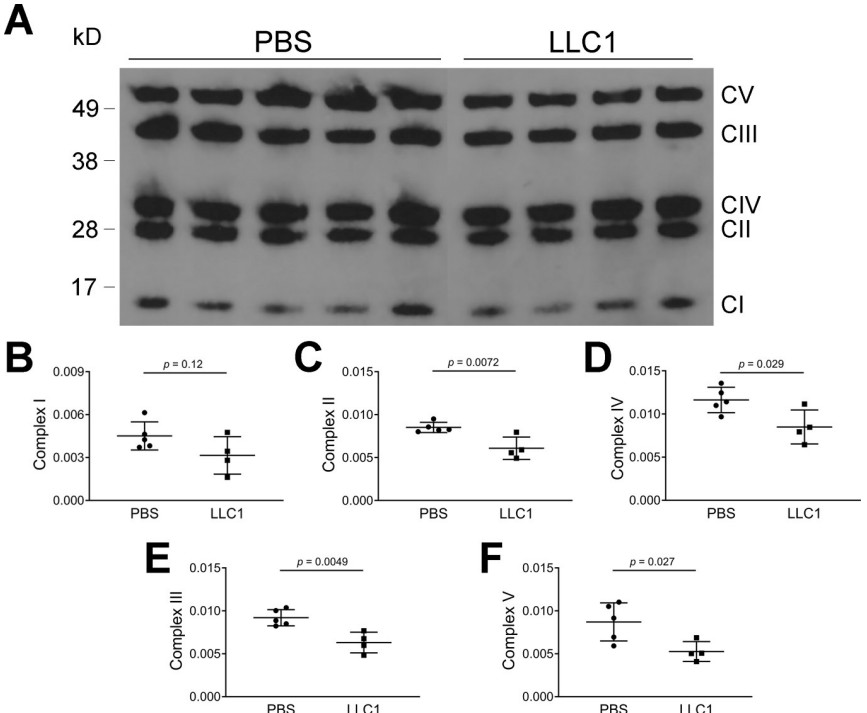

**Fig 6. Effects of LLC1 tumor cells on cardiac mitochondrial OXPHOS complexes.** A. Immunoblot of mitochondrial OXPHOS complexes derived from tumor-injected mice and vehicle-injected mice B. Densitometric analysis of complex I. C. Densitometric analysis of complex II. D. Densitometric analysis of complex IV. E. Densitometric analysis of complex III. F. Densitometric analysis of complex V. Of note, the sequence of densitometric plots corresponds to the position of the complexes on the gel.

## Alterations in cardiac autophagy, proteolysis and apoptosis-related mRNA transcript levels in male tumor-bearing mice

In other rodent tumor models, increased autophagy has been noted in the context of cardiac wasting [13, 33, 34]. To test whether an increased autophagy, proteolysis and apoptosis contribute to a decrease in cardiac wall thickness and cardiac functional impairment in LLC1 tumor-bearing male mice, we measured tissue levels of relevant transcript expression in heart tissues derived from tumor-bearing mice compared to control mice using whole transcriptome sequencing (Tables B-D in S1 File). Changes in some mRNA transcripts of protein influencing autophagy, proteolysis and apoptosis (FDR < 0.05) were observed, but the fold-changes were less than 1.5-fold compared to the transcript levels in control hearts and unlikely to greatly influence cognate protein levels.

## Gene expression in cardiac ventricular tissue derived from male tumor-bearing mice and control mice

To further assess changes in the expression of genes in hearts of tumor-bearing mice that could contribute to reduced cardiac wall thickness and function, we analyzed whole transcriptome data from tumor-bearing mice and control mice. We used a false discovery rate (FDR) of <0.05 and assessed mRNAs that were changed ≤0.5-fold or ≥1.5 fold. Table E in S1 File shows data of the 57 transcripts that were either down-regulated or up-regulated following treatment of mice with LLC1 tumor cells. Notably, apelin (*Apl*) and apelin receptor (*Aplr*) and

*N-Myc*, *Egr1* and *Sox9* mRNAs were significantly decreased. There was a greater than 2 fold increase in the expression of the mRNA for growth arrest and DNA-damage-inducible, β (*Gadd45b*) that encodes a protein involved in the regulation of growth and apoptosis.

### Histological assessment of cardiac fibrosis

No evidence of fibrosis was noted in cardiac tissues derived from tumor-injected mice when analyzed by hematoxylin-eosin or Masson trichrome stains (Figure A in S1 File) shows staining of ventricular tissue from vehicle or tumor-cell injected mice using the Masson trichrome stain).

## Discussion

Cancer incidence is predicted to increase dramatically in the coming decades[3, 35]. Patients with advanced cancer develop cachexia characterized by loss of lean and fat mass and weight loss[36, 37]. This syndrome is associated with increased morbidity and mortality, reduced response to chemotherapy and poor survival following surgery[38–41]. Altered regulation of inflammatory, activin A-myostatin, ubiquitin-proteasome, and autophagy-lysosomal pathways, and mitochondrial dynamics are observed in cancer-associated muscle fiber atrophy and weakness in vivo[42–49]. Tumor cell-derived mediators such as TNFα, IL-6, TWEAK, myostatin and extracellular vesicle HSPs have been postulated to play a role in the pathogenesis of cancer cachexia[42–45, 50–56].

Cancer cachexia involves not only skeletal muscle and fat but also involves several other organs such as the brain, liver, intestine and heart[57]. Reports have described cardiac involvement in patients with cancer even before the initiation of chemotherapy[4, 10]. Indeed, circulating concentrations of cardiovascular biomarker peptides like NT-proBNP, MR-proANP, MR-proADM, CT-pro-ET-1 and hsTnT are increased in an unselected population of patients with cancer prior to induction of any cardiotoxic anticancer therapy and are strongly related to all-cause mortality[10]. These data suggest that the presence of subclinical functional and morphological myocardial damage directly linked to disease progression.

We examined heart function by echocardiography in C57/BL6 mice following the implantation of mouse LLC1 adenocarcinoma cells. Fourteen days after LLC1 implantation, tumor-bearing male mice but not control mice, showed reduced cardiac ejection fraction, and decreased ventricular and septal wall thickness. Of note, the effects of tumor cells on cardiac function were sex-specific as female mice implanted with LLC1 cells failed to exhibit changes in cardiac function. These sex-specific effects of LLC1 tumor cells on heart function are consistent with those found by Cosper and Linewand in CD2F1 mice[20]. Our results showing minor effects of LLC1 cells on cardiac function in female C57BL/6 are similar to those reported by Muhlfeld et al.[58]. The cause for sex-related differences in tumor-induced cardiac dysfunction in our model is not known. Previous investigations have shown that the potent and specific estrogen receptor antagonist, fulvestrant, causes female tumor-bearing mice to lose as much cardiac and body mass as tumor-bearing males[20].

We interrogated potential pathophysiologic processes that are responsible for reduced ventricular wall thickness. We found a significant reduction in protein synthesis rates in ventricular tissue. Since tissue protein is also broken down through proteolysis following ubiquitination, we also measured the mRNA transcript levels of the E3 ubiquitin ligases, RING-finger protein-1 (MuRF1)[59] and F-box only protein 32 (FBXO32) protein[60–63] by whole transcriptome shotgun sequencing (WTSS or RNA-seq). We found minor increases in the mRNA levels of these ubiquitin ligases in heart muscle from tumor-bearing mice. Similarly mRNA transcripts for markers of autophagy, were changed to a minor extent in hearts from tumor-

bearing mice[64–66]. Likewise, minor decreases or increases in mRNA transcripts for apoptosis-related proteins were observed. Thus, in our model of tumor-associated cardiac dysfunction, the changes in ventricular wall thickness in tumor-bearing mice are associated with a reduction in protein synthesis. Increases in protein breakdown and apoptosis are comparatively modest compared to those found in other animal models of cancer stem cell induced cardiac dysfunction. These findings of marked reductions in protein synthesis with relatively modest changes in markers of proteolysis and apoptosis are in contrast to those found in other pre-clinical models of cancer induced heart failure.

Since mitochondria provide energy for muscular contraction in heart and skeletal muscle, we assessed cardiac mitochondrial morphology as well as mitochondrial function. It is known that changes in mitochondrial morphology are associated with alterations in mitochondrial energy production[67]. Cardiac mitochondria from tumor-bearing mice were irregular in ventricular tissue. Measurement of the aspect ratio showed elongation of mitochondria along their longitudinal axes; form factor analysis showed increased mitochondrial branching. These changes in mitochondrial shape were not associated with changes in transcripts of fusion (*Opa1*, *Mfn 1/2*) and fission (*Fis1*, *Drp1*) mediators based on RNA-seq analysis of cardiac tissue. The data suggest that the alterations in mitochondrial shape and branching in cardiac mitochondria from tumor-bearing mice are independent of mRNA expression of fission and fusion mediators. The presence of altered mitochondrial function in the heart was confirmed in AC16 cells where an inhibitory effect of tumor CM on mitochondrial maximal oxygen uptake was noted. Further analysis of mitochondrial function was performed by immunoblot analysis of mitochondrial OXPHOS complex components where statistically significant decreases in expression of complex II-V were noted (Fig 6). However, analysis of the expression of mRNAs for mitochondrial respiratory chain complex components demonstrates slight increases in the expression of mRNAs for various complex components (Table A in S1 File). In light of the reductions in cardiac muscle protein synthesis rates, it is tempting to speculate that the transcriptional message for mitochondrial OXPHOS complex biogenesis may be blocked at the level of protein translation in the presence of LLC1 tumor burden. In aggregate, the data point to the presence of reduced mitochondrial energy production in tumor-bearing mice as a result of reduced mitochondrial oxidation. To our knowledge, previous studies have not demonstrated functional, morphological and biochemical defects in cardiac mitochondria in the context of tumor-induced heart failure.

RNA-seq analysis reveals down regulation of apelin and apelin receptor mRNAs. Apelin causes nitric oxide-dependent vasodilatation, reduces ventricular preload and afterload, and increases cardiac contractility in rats with normal and failing hearts[68, 69]. A decrease in apelin and its receptor could contribute to impaired cardiac function in the heart of tumor-bearing mice. The downregulation of the *N-Myc*, *Egr1*, *Sox9* mRNAs and the upregulation of *Gadd45b* mRNA in tumor-bearing mouse heart could conceivably be important in reducing cardiac growth and differentiation[70, 71]. The down regulation of apelin and the apelin receptor mRNA is also a novel finding and has not been reported in previous studies. It is likely to be of importance because of the reported increases in cardiac contractility induced by apelin. Finally, the down regulation of the *Egr1* may be of significant importance given the role of Egr1 in the maintenance of cellular growth.

In summary, our data show gender-specific reductions in cardiac function, ventricular wall thickness, protein synthesis, mitochondrial oxygen consumption in the presence of LLC1 tumors or conditioned media. Isolation of the factor responsible for reduced protein synthesis and oxygen consumption will yield information about factors altering cardiac function in cancer.

## Supporting information

**S1 File.**
(DOCX)

**S1 Raw Images.**
(PDF)

## Acknowledgments

We acknowledge the assistance of Ms. Katrina Tollefsrud and Mr. Ronald May for the mouse echocardiography and the Mayo Gene Expression and EM Core Facilities. Dr. Joseph Grande of the Department of Laboratory Medicine and Pathology assessed tissues for the presence of fibrosis.

## Author Contributions

**Conceptualization:** Jessica M. Dorschner, Ian R. Lanza, Rajiv Kumar.

**Data curation:** Taylor E. Berent, Jessica M. Dorschner, Thomas Meyer, Theodore A. Craig, Xuewei Wang, Hawley Kunz.

**Formal analysis:** Jessica M. Dorschner, Thomas Meyer, Xuewei Wang, Hawley Kunz, Ian R. Lanza, Horng Chen, Rajiv Kumar.

**Funding acquisition:** Rajiv Kumar.

**Investigation:** Taylor E. Berent, Jessica M. Dorschner, Thomas Meyer, Theodore A. Craig, Xuewei Wang, Hawley Kunz, Aminah Jatoi, Ian R. Lanza, Horng Chen, Rajiv Kumar.

**Methodology:** Jessica M. Dorschner, Rajiv Kumar.

**Writing – original draft:** Aminah Jatoi, Ian R. Lanza, Horng Chen, Rajiv Kumar.

**Writing – review & editing:** Taylor E. Berent, Jessica M. Dorschner, Thomas Meyer, Theodore A. Craig, Xuewei Wang, Hawley Kunz, Aminah Jatoi, Ian R. Lanza, Horng Chen, Rajiv Kumar.

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
