## [Decision Letter · Decision Letter 0]

18 Sep 2019

PONE-D-19-19216

Impaired cardiac performance, protein synthesis, and mitochondrial function in tumor-bearing mice

PLOS ONE

Dear Dr. Kumar,

Thank you for submitting your manuscript to PLOS ONE. After careful consideration, we feel that it has merit but does not fully meet PLOS ONE’s publication criteria as it currently stands. Therefore, we invite you to submit a revised version of the manuscript that addresses the points raised during the review process.

The reviewers point out an issue with the novelty, the lack of depth of the mechanistic studies  and the unconvincing RNA-seq data. The editor concurs. Please perform further studies to provide more insightful information to address the comments as suggested.

We would appreciate receiving your revised manuscript by Nov 02 2019 11:59PM. To enhance the reproducibility of your results, we recommend that if applicable you deposit your laboratory protocols in protocols.io, where a protocol can be assigned its own identifier (DOI) such that it can be cited independently in the future. For instructions see: http://journals.plos.org/plosone/s/submission-guidelines#loc-laboratory-protocols

We look forward to receiving your revised manuscript.

Kind regards,

Xiongwen Chen, PhD

Academic Editor

PLOS ONE

Journal Requirements:

2. In the Methods, please include the total number of mice used in this study.

Additional Editor Comments (if provided):

The reviewers point out an issue with the novelty, the lack of depth of the mechanistic studies and the unconvincing RNA-seq data. The editor concurs. Therefore, please address the comments with further studies as suggested.

Reviewers' comments:

Reviewer's Responses to Questions

**Comments to the Author**

1. Is the manuscript technically sound, and do the data support the conclusions?

Reviewer #1: Yes

Reviewer #2: Partly

2. Has the statistical analysis been performed appropriately and rigorously? 

Reviewer #1: Yes

Reviewer #2: Yes

3. Have the authors made all data underlying the findings in their manuscript fully available?

Reviewer #1: No

Reviewer #2: Yes

4. Is the manuscript presented in an intelligible fashion and written in standard English?

Reviewer #1: Yes

Reviewer #2: Yes

5. Review Comments to the Author

Reviewer #1: In this article, Dorschner et al., investigated the underlying mechanisms of cancer –induced cardiac dysfunction using a tumor-bearing mouse model. Dorschner et al. observed that male tumor-bearing mice manifest the reduced ejection fraction and ventricular wall thickness. However, this phenomena does not occur in female tumor-bearing mice. Dorschner et al. further hypothesize that reduced ventricular wall thickness and ejection fraction observed in male tumor-bearing mice could be associated with impaired protein synthesis and mitochondria function. Evidence showed that protein synthesis is reduced and mitochondria morphology and function are changed as well in male tumor-bearing mice. Dorschner et al. also performed RNA-seq and found differentially changed genes are associated with protein synthesis as well as mitochondria function. Therefore, Dorschner et al. concluded that cardiac atrophy in male tumor-bearing mice is associated with reduced protein synthesis and impaired contractile function is related to altered mitochondria function. Several comments are listed below

1. There is no detailed discussion why authors observed cancer-induced cardiac dysfunction in male, but not in female

2. By looking at the RNA-seq data, actually, almost all genes listed in the tables are not significantly changed (<1.5), which is not convincing that these subtle changes are associated with cardiac dysfunction. I will suggest authors also test the protein levels with immunoblotting using hall mark genes associated with protein synthesis and mitochondria function to validate the RNA-seq data.

3. If possible, pathological data such as fibrosis could be added which would be more supportive for the conclusion

4. The resolution of the figures is not high.

5. Figure 1-4 can be integrated into two figures. Characterization of male mice can be consolidated into figure 1, and female data in figure 2

6. In figure 5, authors used over 9 animals in B and C, but why only 4 animals were used in A?

7. I would suggest moving the tables with next generation sequencing data to supplement.

Reviewer #2: In the present manuscript, Kumar and colleagues aimed to address pathological changes in the heart under tumor-bearing conditions. The authors utilized lung tumor as an example to induce cachexia and evaluated heart function. They showed, in a convincing way, cardiomyopathy emerged at 14 days post tumor inoculation. Moreover, the heart growth was impaired. The authors went further to show mitochondria had certain abnormalities and maximal oxygen consumption was reduced.

1. The studies were thoughtfully designed and nicely executed. However, the conclusions suffer from questions in novelty. Tumor-related cachexia and related heart disease has been noted for years. Moreover, previous studies have suggested the increase in atrophy and decrease in mitochondrial action may contribute. For a few examples, please see, Spring H, 2014, EHJ; Tian M, 2010, International J of Oncology, etc. Therefore, the present study may not provide significant conceptual advances in the field of cachexia-associated heart disorder.

2. In the last figure, conditioned media (CM) from tumor cells inhibited OCR. Is this due to inflammatory cytokines? If so, how to narrow down the candidates?

3. The authors have done thorough analysis for the RNA-Seq data. What about changes at the protein level? Such as atrophy-related proteins and the growth signaling pathway (i.e. mTOR).

4. The authors may need to compare the ratio of heart weight/tibia length, since body weight may be affected by tumor.

6. PLOS authors have the option to publish the peer review history of their article (what does this mean?). If published, this will include your full peer review and any attached files.

Reviewer #1: No

Reviewer #2: No

---

## [Author Response · Author response to Decision Letter 0]

27 Sep 2019

Response to reviewers. 

ALL CHANGES HAVE BEEN UNDERLINED IN THE REVISED MANUSCRIPT.

Reviewer #1: 

Summary of findings: In this article the authors investigated the underlying mechanisms of cancer-induced cardiac dysfunction using a tumor-bearing mouse model. They observed that male tumor-bearing mice manifest the reduced ejection fraction and ventricular wall thickness. However, these phenomena do not occur in female tumor-bearing mice. They further hypothesize that reduced ventricular wall thickness and ejection fraction observed in male tumor-bearing mice could be associated with impaired protein synthesis and mitochondria function. Evidence showed that protein synthesis is reduced and mitochondria morphology and function are changed as well in male tumor-bearing mice. The authors also performed RNA-seq and found differentially changed genes are associated with protein synthesis as well as mitochondria function. Therefore, they concluded that cardiac atrophy in male tumor-bearing mice is associated with reduced protein synthesis and impaired contractile function is related to altered mitochondria function. Several comments are listed below:

Comments and responses:

Comment 1. There is no detailed discussion why authors observed cancer-induced cardiac dysfunction in male, but not in female. 

Response: We discuss why female mice do not respond in a manner similar to male mice. Previous work (Cosper PF, Leinwand LA. Cancer causes cardiac atrophy and autophagy in a sexually dimorphic manner. Cancer Res. 2011;71(5):1710-20) has shown that this is mediated by the estrogen receptor. The use of an estrogen receptor blocking agent, fulvestrant, in female mice with cancer, is associated with a response that is similar to that observed in male mice. We have now added text discussing this issue.

Comment 2. By looking at the RNA-seq data, actually, almost all genes listed in the tables are not significantly changed (<1.5), which is not convincing that these subtle changes are associated with cardiac dysfunction. I will suggest authors also test the protein levels with immunoblotting using hall mark genes associated with protein synthesis and mitochondria function to validate the RNA-seq data.

Response: We now provide immunoblot data showing that changes in the mRNAs for components of the mitochondrial respiratory chain complex are paralleled by changes observed in protein components of the respiratory chain complex assessed by (new figure 6). 

Comment 3. If possible, pathological data such as fibrosis could be added which would be more supportive for the conclusion

Response: We examined cardiac tissues by histology and did not notice an increase in fibrosis. A statement to this effect is now included in the manuscript.

Comment 4. The resolution of the figures is not high.

Response: This has now been corrected. 

Comment 5. Figure 1-4 can be integrated into two figures. Characterization of male mice can be consolidated into figure 1, and female data in figure 2

Response: This has now been done. 

Comment 6. In figure 5, authors used over 9 animals in B and C, but why only 4 animals were used in A?

Response: We had only 4 samples available to us for analysis of 13C phenylalanine incorporation into protein. 

Comment 7. I would suggest moving the tables with next generation sequencing data to supplement.

Response: We have moved the next generation sequencing data into the supplement. 

Reviewer #2

Summary of findings: In the present manuscript, Kumar and colleagues aimed to address pathological changes in the heart under tumor-bearing conditions. The authors utilized lung tumor as an example to induce cachexia and evaluated heart function. They showed, in a convincing way, cardiomyopathy emerged at 14 days post tumor inoculation. Moreover, the heart growth was impaired. The authors went further to show mitochondria had certain abnormalities and maximal oxygen consumption was reduced.

Comment 1. The studies were thoughtfully designed and nicely executed. However, the conclusions suffer from questions in novelty. Tumor-related cachexia and related heart disease has been noted for years. Moreover, previous studies have suggested the increase in atrophy and decrease in mitochondrial action may contribute. For a few examples, please see, Spring H, 2014, EHJ; Tian M, 2010, International J of Oncology, etc. Therefore, the present study may not provide significant conceptual advances in the field of cachexia-associated heart disorder.

Response: Previous studies have demonstrated heart failure in various animal models injected with different types of tumor cells. Our study, however, significantly adds to knowledge about mechanisms responsible for reduced cardiac function. First, to our knowledge, we are unaware of studies performed previously with LLC1 cells, a model of lung non-small cell tumors. Second, we are unaware of studies in which investigators have examined the effect of tumor conditioned medium on mitochondrial oxygen uptake and the effect of tumor cell implantation on cardiac protein synthesis. Third, using whole transcriptome sequencing, we show reductions in the expression of mRNAs which encode proteins that affect cardiac function and growth e.g. Apelin, Apelin receptor, Sox 9, and Egr1. Increases in the expression of mRNAs involved in the regulation of growth arrest and apoptosis such as Gadd45B is noted in tumor-bearing mice. 

Comment 2. In the last figure, conditioned media (CM) from tumor cells inhibited OCR. Is this due to inflammatory cytokines? If so, how to narrow down the candidates?

Response: We thank the reviewer for the comment. We do not know the mediator for a reduction in mitochondrial oxygen consumption rate. This is an ongoing project in our laboratory at the present time. We have added text to address this issue and potential paths that can be taken to isolate such a mediator.

Comment 3. The authors have done thorough analysis for the RNA-Seq data. What about changes at the protein level? Such as atrophy-related proteins and the growth signaling pathway (i.e. mTOR).

Response: Please see the response to comment 2 by reviewer 1. We now provide data showing that changes in the mRNAs for components of the mitochondrial respiratory chain complex are paralleled by changes observed in components of the respiratory chain complex assessed by Western blotting (new figure 6). 

Comment 4. The authors may need to compare the ratio of heart weight/tibia length, since body weight may be affected by tumor.

Response: We thank the reviewer for the comment. Unfortunately, animal carcasses are no longer available for us to make this determination.

---

## [Decision Letter · Decision Letter 1]

8 Nov 2019

PONE-D-19-19216R1

Impaired cardiac performance, protein synthesis, and mitochondrial function in tumor-bearing mice

PLOS ONE

Dear Dr. Kumar,

Thank you for submitting your manuscript to PLOS ONE. After careful consideration, we feel that it has merit but does not fully meet PLOS ONE’s publication criteria as it currently stands. Therefore, we invite you to submit a revised version of the manuscript that addresses the points raised during the review process.

These two external reviewers are satisfied with your revision and only a minor addition of evidence of fibrosis is requested. Please revise accordingly and the revised version will expedited for acceptance.

We would appreciate receiving your revised manuscript by Dec 23 2019 11:59PM. To enhance the reproducibility of your results, we recommend that if applicable you deposit your laboratory protocols in protocols.io, where a protocol can be assigned its own identifier (DOI) such that it can be cited independently in the future. For instructions see: http://journals.plos.org/plosone/s/submission-guidelines#loc-laboratory-protocols

We look forward to receiving your revised manuscript.

Kind regards,

Xiongwen Chen, PhD

Academic Editor

PLOS ONE

Reviewers' comments:

Reviewer's Responses to Questions

**Comments to the Author**

1. If the authors have adequately addressed your comments raised in a previous round of review and you feel that this manuscript is now acceptable for publication, you may indicate that here to bypass the “Comments to the Author” section, enter your conflict of interest statement in the “Confidential to Editor” section, and submit your "Accept" recommendation.

Reviewer #1: All comments have been addressed

Reviewer #2: All comments have been addressed

2. Is the manuscript technically sound, and do the data support the conclusions?

Reviewer #1: Partly

Reviewer #2: Yes

3. Has the statistical analysis been performed appropriately and rigorously? 

Reviewer #1: Yes

Reviewer #2: Yes

4. Have the authors made all data underlying the findings in their manuscript fully available?

Reviewer #1: Yes

Reviewer #2: Yes

5. Is the manuscript presented in an intelligible fashion and written in standard English?

Reviewer #1: Yes

Reviewer #2: Yes

6. Review Comments to the Author

Reviewer #1: The authors have adequately addressed my concerns. I appreciate that the authors added the statement of cardiac fibrosis, even though the authors didn't observe any significant difference, I would suggest the authors add the figure evidence to the supplement.

Reviewer #2: I want to thank the authors for addressing my concerns on technical issues. While I agree with the authors that this study provides certain novel insights about this type of tumor cells in triggering heart disease, the authors may consider to discuss previous findings in a manner of comparing with the current work.

7. PLOS authors have the option to publish the peer review history of their article (what does this mean?). If published, this will include your full peer review and any attached files.

Reviewer #1: No

Reviewer #2: No

---

## [Author Response · Author response to Decision Letter 1]

13 Nov 2019

RESPONSE TO REVIEWERS. 

ALL CHANGES HAVE BEEN HIGHLIGHTED IN YELLOW IN THE REVISED MANUSCRIPT.

Reviewer #1: 

Comments and responses:

Comment 1. The authors have a adequately addressed my concerns. I appreciate that the author is added the statement of cardiac fibrosis, even though the author is did not observe any significant difference, I would suggest the author is added the figure evidence to the supplement.

Response: We have now added of figure (Supplemental Figure 1) showing the absence of fibrosis in cardiac tissue derived from tumor-implanted mice using the Masson Trichrome which is exceptionally sensitive for sensitive for fibrosis. As noted, we did not observe fibrosis in either the vehicle or tumor cell injected mice.

Reviewer #2

Comment 1. I want to thank the authors for addressing my concerns on technical issues. While I agree with the authors that this study provides certain novel insights about this type of tumor cells in triggering heart disease, the authors may consider discussing previous findings in a matter of comparing with current work

Response: Thank you for your comments. We have added sentences now in which we discuss our work in comparison to work previously published by others.

---

## [Decision Letter · Decision Letter 2]

27 Nov 2019

Impaired cardiac performance, protein synthesis, and mitochondrial function in tumor-bearing mice

PONE-D-19-19216R2

Dear Dr. Kumar,

We are pleased to inform you that your manuscript has been judged scientifically suitable for publication and will be formally accepted for publication once it complies with all outstanding technical requirements.

With kind regards,

Xiongwen Chen, PhD

Academic Editor

PLOS ONE

Additional Editor Comments (optional):

Reviewers' comments:

Reviewer's Responses to Questions

**Comments to the Author**

1. If the authors have adequately addressed your comments raised in a previous round of review and you feel that this manuscript is now acceptable for publication, you may indicate that here to bypass the “Comments to the Author” section, enter your conflict of interest statement in the “Confidential to Editor” section, and submit your "Accept" recommendation.

Reviewer #1: All comments have been addressed

Reviewer #2: All comments have been addressed

2. Is the manuscript technically sound, and do the data support the conclusions?

Reviewer #1: Yes

Reviewer #2: Yes

3. Has the statistical analysis been performed appropriately and rigorously? 

Reviewer #1: Yes

Reviewer #2: Yes

4. Have the authors made all data underlying the findings in their manuscript fully available?

Reviewer #1: Yes

Reviewer #2: Yes

5. Is the manuscript presented in an intelligible fashion and written in standard English?

Reviewer #1: Yes

Reviewer #2: Yes

6. Review Comments to the Author

Reviewer #1: The authors have addressed all my questions and I have No further concerns.

Reviewer #2: No further revision is needed. All questions have been satisfactorily addressed. I want to thank the authors for addressing the concerns.

7. PLOS authors have the option to publish the peer review history of their article (what does this mean?). If published, this will include your full peer review and any attached files.

Reviewer #1: No

Reviewer #2: No

---

## [Editor Report · Acceptance letter]

10 Dec 2019

PONE-D-19-19216R2 

Impaired cardiac performance, protein synthesis, and mitochondrial function in tumor-bearing mice 

Dear Dr. Kumar:

I am pleased to inform you that your manuscript has been deemed suitable for publication in PLOS ONE. Congratulations! Your manuscript is now with our production department. 

With kind regards,

on behalf of

Dr Xiongwen Chen 

Academic Editor

PLOS ONE